Web party effect: a cocktail party effect in the web environment

Rigutti Sara 1
Fantoni Carlo 1 2 cfantoni@units.it
Gerbino Walter 1
1 Department of Life Sciences, Psychology Unit “Gaetano Kanizsa,” University of Trieste , Trieste , Italy
2 Center for Neuroscience and Cognitive Systems@UniTn, Italian Institute of Technology , Rovereto , Italy
Macknik Stephen
Electronic publication date: 2015 Mar 10
Publication date: 2015
Volume: 3
Electronic Location ID: e828
Received 2014 Dec 24; Accepted 2015 Feb 18
Copyright: © 2015 Rigutti et al.
Copyright year: 2015
Copyright holder: Rigutti et al.
License: This is an open access article distributed under the terms of the Creative Commons Attribution License, which permits unrestricted use, distribution, reproduction and adaptation in any medium and for any purpose provided that it is properly attributed. For attribution, the original author(s), title, publication source (PeerJ) and either DOI or URL of the article must be cited.
License URL: https://creativecommons.org/licenses/by/4.0/

Keywords: Attention, Web navigation, Information seeking, Web layout, Search time, Usability, Cognitive load, Experienced duration, Complexity, Mediation

Funding: European Social Fund 2007/13 S.H.A.R.M Italian Ministry economic development University of Trieste FRA-2013 This work was supported by a grant of the European Social Fund 2007/13—S.H.A.R.M. project—awarded to Sara Rigutti (PI Walter Gerbino). Partial support was also provided by the Italian Ministry of Economic Development (Industria 2015, Ecoautobus Grant to Walter Gerbino) and by the University of Trieste (FRA-2013 Grant to Carlo Fantoni). The funders had no role in study design, data collection and analysis, decision to publish, or preparation of the manuscript.

==============================
In goal-directed web navigation, labels compete for selection: this process often involves knowledge integration and requires selective attention to manage the dizziness of web layouts. Here we ask whether the competition for selection depends on all web navigation options or only on those options that are more likely to be useful for information seeking, and provide evidence in favor of the latter alternative. Participants in our experiment navigated a representative set of real websites of variable complexity, in order to reach an information goal located two clicks away from the starting home page. The time needed to reach the goal was accounted for by a novel measure of home page complexity based on a part of (not all) web options: the number of links embedded within web navigation elements weighted by the number and type of embedding elements. Our measure fully mediated the effect of several standard complexity metrics (the overall number of links, words, images, graphical regions, the JPEG file size of home page screenshots) on information seeking time and usability ratings. Furthermore, it predicted the cognitive demand of web navigation, as revealed by the duration judgment ratio (i.e., the ratio of subjective to objective duration of information search). Results demonstrate that focusing on relevant links while ignoring other web objects optimizes the deployment of attentional resources necessary to navigation. This is in line with a web party effect (i.e., a cocktail party effect in the web environment): users tune into web elements that are relevant for the achievement of their navigation goals and tune out all others.

Introduction

The ability to tune into a single voice and tune out all others during a crowded party is known as the cocktail party effect (Cherry, 1953). This effect demonstrates the selectivity of human mechanisms that allow us “to attend”; i.e., to prepare ourselves for the extraction of the relevant information from a cluttered environment (Broadbent, 1958; Treisman, 1964; Deutsch & Deutsch, 1963; Johnston & Heinz, 1978). The peculiar human ability to successfully focus and shift attention to optimize information extraction is ascribable to a rather general process controlling both visual and auditory inputs (Shinn-Cunningham, 2008). A visual analogue of the cocktail party effect can indeed be found in explicit attentional capture phenomena (i.e., inattentional blindness) demonstrating a surprising degree of blindness to salient visual features (e.g., dynamic information; Simons & Chabris, 1999; Simons, 2000).

Also, navigation within web pages has several close relatives with the cocktail party effect; it is indeed an information seeking activity performed within a cluttered environment, in which all web options competing for selection (i.e., links) behave as those noisy voices at the cocktail party. In the case of web navigation, therefore, the achievement of an information goal and the time required to achieve it (i.e., the information seeking time) depend on the capacity of the user to tune into those web elements that are relevant for the achievement of the goal and tune out all others. Consistently with such a commonality here we ask whether a similar effect of selective attention might occur during web navigation within ordinary websites; hence, we ask whether a web party, like a cocktail party, effect exists. During goal-directed web navigation, is the competition for web selection entered by all navigation options or by only those options that are more likely to be functional to information seeking?

In this respect one can distinguish between navigation options embedded in conventional elements that, according to internalized layout conventions, are more likely to be functional to information seeking (e.g., navigation bar, footer, menu categories; Nielsen, 1999), and less conventional web options (e.g., images, banners, embedded links, words). According to a web party effect, the information seeking time within the web should be moulded by the former types of options but not the latter.

Our study corroborated this hypothesis by: (1) accounting for the role of selective attention in goal-directed navigation using an information seeking task within a representative set of websites of Small and Medium Enterprises (SME) 1; and (2) contrasting the predictive power of a novel measure of home page complexity based on a part of web options, against several other candidate predictors of information seeking time that have been proposed so far as valuable indices of web page complexity (e.g., the overall number of links, words, images, and graphical regions, or the JPEG file size of the homepage screenshot).

Web information seeking behavior

Our idea reflects previous alternative approaches to the modeling of web information seeking behavior: semantic- vs. knowledge-based approaches.

According to Pirolli & Fu (2003; but see also Chi, Pirolli & Pitkow, 2000; Card et al., 2001; Katz & Byrne, 2003; Kitajima, 2003; Miller & Remington, 2004; Blackmon, Kitajima & Polson, 2005; Kaur & Hornof, 2005) web information seeking behavior is a matter of information scent (Pirolli & Card, 1999), which is grounded on semantic rather than procedural knowledge, including an estimate of the relevance of all links within a web page. CoLiDeS (Comprehension-based Linked model of Deliberate Search; Kitajima, Blackmon & Polson, 2005) takes a step further; it assigns information scent for a user goal to not only the links but also the sub-regions within a web page, according to the users’ prior knowledge of layout conventions. Users select one patch of information within the web page and ignore all the rest, through a mechanism of selective attention optimizing attentional demand. Real websites, indeed, are not composed of labels and textual materials only, but also by different types of web objects (navigation bars, canvas, footers, embedded links) placed in different page regions (top, bottom, central, left, right) as used to display, group, and emphasize the relevance of labels.

Several studies have shown that an organization of web page layout reflecting the hierarchical structure of information contents (Rosenfeld & Morville, 1998; Veen, 2000), helps information seeking behavior; it reduces information seeking time and improves information search accuracy (Tullis, 1998; Bernard, Hull & Drake, 2001; Pearson & Van Schaik, 2003; Halverson & Hornof, 2004; Ling & Van Schaik, 2004; Rigutti & Gerbino, 2004). Selection time has been found to increase as the target label deviates farther from the top-left region of the web page (Van Schaik & Ling, 2001; Rigutti, Fantoni & Gerbino, 2007; Tamborello II & Byrne, 2007; Rigutti, Gerbino & Fantoni, 2008). Eye movements during web page scanning follow the reading direction from left-to-right and/or top-to-bottom paths with a dominance of top and middle areas of the web page (Faraday, 2000; Goldberg et al., 2002; Joachims et al., 2005; Nielsen, 2006; Buscher, Cutrell & Morris, 2009). Evidence from the observation of how users interact with the web browser in the working environment over a long period provides further support for the relevance of web page layout as well as of users’ prior knowledge about layout conventions (Catledge & Pitkow, 1995; Cockburn & McKenzie, 2001; Weinreich et al., 2006). Analyzing the clickstream Weinreich et al. (2006) found that users did not take the time to read every page completely, but just glimpsed over most of the information offered, by privileging links laying in the top/left region of the page that are embedded within the principal navigation elements (i.e., horizontal and vertical navigation bars).

Web page complexity

Despite these different perspectives on web information seeking behavior, scientists agree to consider web page complexity as a valuable predictor of information seeking time, which is the focus of the current study. Gwizdka & Spence (2006) defined the complexity of a web page in term of the overall number of links within the page (i.e., navigation choices). Several studies corroborated the predictive power of such a definition by finding that increasing the number of links per page decreases link selection accuracy and increases information seeking time (Pierce, Sisson & Parkinson, 1992; Mosenthal, 1996; Blackmon et al., 2002; Usher, Olami & McClelland, 2002; Van Schaik & Ling, 2012). As an alternative measure, recent works have proposed the number of bytes of the JPEG screenshot as a valuable predictor of users’ overall experience of pleasantness, psychophysiological responses as heart rate and electrodermal activity, performance (i.e., information seeking time), and memory (Tuch et al., 2009; Stickel, Ebner & Holzinger, 2010). Other works (Ivory, Sinha & Hearst, 2000; Ivory, Sinha & Hearst, 2001; Michailidou, Harper & Bechhofer, 2008; Harper, Michailidou & Stevens, 2009; Miniukovich & De Angeli, 2014) provided compound measures of web page complexity based on different weighted linear combinations of these and other types of metrics such as the number of words, images, and segregated graphical regions.

A knowledge-based approach to web information seeking behavior as formalized by CoLiDes sets the basis for an alternative hypothesis: information seeking time is accounted for by the complexity of the home page, defined by those only options and links which are more likely to be functional for information seeking, rather than by the entire set of web features which are detectable within the page (i.e., number of words, images, links, segregated graphical regions).

Rationale and expectations: a knowledge-based measure of complexity

As the number of relevant links within a page generally covaries with most of the above mentioned web page complexity metrics, there is no up-to-date evidence allowing to establish whether information seeking time should be better predicted by the encoding of all or just part of the information within the home page. Here we answer such a question by testing a new measure of web page complexity (C) that formalizes three basic constraints of a knowledge-based approach to web information seeking behavior: (1) only links embedded within relevant web elements are encoded and contribute to web page complexity; (2) their number is multiplied by a composite factor that takes into account both the number of relevant web elements within the web page and (3) the weights of web elements they belong to. Such constraints are formalized by the following equation: (1) C=NL∑i=1nEiωi

where NL stands for the number of links embedded in relevant web elements, i is the web element index between n = 1 and n = 5 (horizontal navigation bar, left vertical menu, right vertical menu, menu within the canvas, footer), Ei is a dichotomous variable (1 for presence and 0 for absence of the element), and ωi is the weight of the ith element in the [0, ∞] range.

According to state-of-the-art research on selection time within hierarchical menu trees (Lee & MacGregor, 1985), we assumed that information seeking time (IST) increases as C increases, as follows: (2) IST=αC+θ

where α formalizes the individual search strategy, from exhaustive to self-terminating (Paap & Roske-Hofstrand, 1986), and θ stands for the individual latency of the action required by the task.

Figure 1 shows the predicted complexity of home pages for different combinations of web elements and links, calculated according to our set of empirically extracted weights (see ‘Data analysis’ subsection for details). Each combination defines a line with origin in [NL = 0, C = 0] and a slope equal to the factor that sums up the weights of navigation elements present in the home page.

Figure 1 Predicted complexity (C) as a function of the number of links for different combinations of web navigation elements.

In (A) the color of the lines codes different combinations of navigation elements as specified in the legend on top. The three dots refer to representative home pages used in our experimental sample and discussed in the text. (B) Distribution of complexity scores for the home pages of 1945 Friuli Venezia Giulia SME sites, from which our representative sample of home pages has been extracted. ATECO categories are color-coded (see the ‘Stimuli and apparatus’ subsection for details).

Note that the model tolerates predictions that are both consistent and inconsistent with a semantic-based approach to web information seeking behavior. Consistently with a semantic-based approach, predicted complexity increases as NL grows larger (C of page a smaller than C of page c). However, the model also includes strongly counterintuitive predictions like the following: page c, with a small number of links and navigation elements, scores a higher complexity than page b, with a larger number of links and navigation elements. The numerous links embedded within the right bar of page b poorly contribute to the overall complexity score, while those few links embedded in the central categories of the page (i.e., canvas) c have a high impact on the overall complexity score. As a result, the complexity scores for pages c and b, as well as their IST, are similar (with c slightly larger than b) despite large differences in the number of selection choices and graphical elements.

Our experiment considers and tests the following four hypotheses.

(H1) The time required to search for an information goal located at a given depth (starting from the home page) does not depend on the number of links (NL) or web elements (NE) considered in isolation, but rather on their weighted combination as formalized by C in Eq. (1): independent and combined effects of NL and NE on IST should be entirely accounted for by our synthetic C measure.

(H2) Consistently with a web party effect, our C measure should explain a larger amount of variance of the IST distribution than alternative metrics of web page complexity, which are not intended to formalize selective attention processing of the page (e.g., overall number of links, bytes, words, images, segregated graphical regions).

(H3) C is expected to be negatively correlated with our explicit measure of website usability (EU), given that one major goal of web designers is the making of usable and beautiful graphical user interfaces while reducing artifact complexity.

(H4) C is expected to be diagnostic of the cognitive effort involved in a web information search task. According to the numerous studies on the distortions of time perception produced by cognitive load (Eisler, 1976; Zakay, Nitzan & Glicksohn, 1983; Meyer et al., 1996), we expect C to be inversely correlated with the ratio between the estimated duration of the information search (i.e., Estimated Search Time, EST) and IST. This expectation arises from predictions of attentional models of experienced duration (Thomas & Weaver, 1975; Zakay & Block, 1995; Zakay & Block, 1996); a proportional decrement in duration estimates for increasing cognitive effort is indeed predicted when a prospective duration paradigm is used, like in our study in which participants were informed in advance that they should express a duration judgment.

Method

We tested our expectations by measuring IST, EST, and EU for a representative set of home pages extracted from the population of SME sites of our regional district (Regione Autonoma Friuli Venezia Giulia, FVG). The visual complexity of the starting home page of selected sites displayed a large variability. Every participant was asked first to search for an information goal located two clicks away from the home page, then to explicitly estimate the duration of information seeking navigation and to evaluate his/her global view of website usability on the System Usability Scale (SUS).

Participants

Twenty undergraduates of the University of Trieste participated in the experiment. The sample consisted of 8 males (M 26.5 years, SD 7.7) and 12 females (M 24.3 years, SD 4.2). All had normal or corrected-to-normal vision and were naive as to the purpose of the experiment. We administered a “user profile” questionnaire at the beginning of the experimental session to obtain general user data regarding web experience and typical online behavior patterns. All but two participants reported more than five years of general computer usage, and all but one reported having used the web at least two hours per week for more than one year. Regarding current web usage, all but two participants reported at least five hours per week. Sixteen participants reported purchasing items online in the past, with eleven reporting having purchased at least five items online and all of them using regularly the email.

The study was approved by the Research Ethics Committee of the University of Trieste (approval number 52) in compliance with national legislation, the Ethical Code of the Italian Association of Psychology, and the Code of Ethical Principles for Medical Research Involving Human Subjects of the World Medical Association (Declaration of Helsinki). All participants provided their written informed consent prior to inclusion in the study, accepted the response sheet of the SUS questionnaire used at the end of the navigation session, and therefore behaved as active participants in the entire data collection. Response sheets were filed as raw documents.

Stimuli and apparatus

The experiment took place in the Active Vision Laboratory of the Department of Life Sciences of the University of Trieste. The participant was seated in front of a 21″ LCD Computer Monitor (Sony Trinitron Color Graphic Display GDM-F520 1280 × 768 pixels), at a comfortable viewing distance from the screen (about 60 cm), in a dimly lit room. The participant controlled all the progress of the experiment using a computer mouse with his/her right hand. To avoid problems due to the variability of web page load time and, more in general, to online navigation (e.g., server crash, slowing down of TCP tuning protocol) all navigation sessions were conducted off-line. The entire set of 26 websites was downloaded through HTTrack Website Copier in a local directory and the Xampp platform as supported by the Apache HTTP Server was customized so to trace logfiles and store the navigation paths as well as the timing associated to each web page click with millisecond precision. Each website was displayed and navigated using Google Chrome browser. Real websites were used to assure a high ecological validity of the experiment. Based on a previous analysis of the entire FVG SME website set (N = 1945), we selected 26 representative websites, so chosen to cover a good spread of standard visual types normally encountered within the FVG SME website set (Fig. 1B). The selection of home pages of live sites was based on a 3-level classification (simple, intermediate, and complex), according to general principles of web usability design and considering standard metrics of home page complexity: number of words (M = 227.6, range = 32–756, SD = 176.9), number of segregated graphical regions computed by closely following the rationale for chunk rendering evaluation described in Harper, Michailidou & Stevens (2009) (M = 9.15, range = 1–19, SD = 4.4), number of links (M = 27.65, range = 6–84, SD = 17.6), number of web navigation elements counted amongst four main types—horizontal navigation bar, vertical menus, menu in the canvas, and footer—(M = 2.1, range = 1–3, SD = 0.8), number of links embedded within the web navigation elements (M = 16.96, range = 6–56, SD = 10.5), and number of dynamic and static images (M = 14.43, range = 2–33, SD = 7.9), JPEG file size of the 1280 × 768 screenshots of the portion of the home page viewable at first sight, computed by closely following the method described in Tuch et al. (2009) (M = 240 byte, range = 154–391, SD = 57.5).

Experimental websites were selected so as to be representative of local SME sectors according to the ATECO 2007 economical categorization (Official Journal, 20 December 2006, CE n.1893/2006, 20/12/2006) as provided by the Trieste’s Chamber of Commerce. We categorized the subset of 26 experimental websites into the following types, corresponding to the five most numerous categories within the entire FVG SME website set (as shown by percentages within parentheses): hotels/restaurants (N = 1, 10%), constructions (N = 2, 12%), commerce (N = 3, 13%), services (e.g., health, ICT, education; N = 7, 27%), factories/manufactures (N = 14, 38%). The function relating the number of experimental sites per ATECO category to the percentage of sites per ATECO category (within the entire population of sites to which participants have been likely exposed) was close to linearity (slope = 0.43, intercept = − 3.27, r2 = 0.97). Such a linear relation guarantees for the ecological validity of our experimental set and minimizes possible systematic biases due to an unbalance between individual familiarity with different site categories and their presence/absence within the experimental set.

To keep the structural complexity of the task constant, we selected all information goals two clicks away from the home page (a depth constraint related to information architecture) and all links to the page displaying the information goal in the central region of the intermediate page measuring 500 px horizontally × 360 px vertically (a visuo-spatial constraint): 12% of target links in the intermediate page belonged to the secondary horizontal navigation bar (displayed below the main horizontal navigation bar), 31% to the left vertical navigation bar, and 58% to the categories displayed in the central region of the page.

In the home page, we avoided possible biasing effects of link visibility by selecting target items included in the page portion displayed at the onset, without scrolling (Mx = 477 px; SDx = 249; My = 187 px; SDy = 156): 65% of target items belonged to the horizontal navigation bar placed in the top region of the page and 35% to the vertical navigation bar placed in the top left side of the page. Furthermore, all target items were superordinate meaning words (38% “products”; 27% among “catalogue,” “collections,” “systems”; 23% “services”; 11% among “rooms,” “calendars,” “industrial machinery”), as needed to balance their semantic access.

The Italian translation of the SUS (Argentero et al., 2009) was administered at the end of each navigation trial. The response form contained the 10 SUS items each flanked on the right side by the 5-point agreement scale numbered and ordered from 1 (strongly disagree) to 5 (strongly agree).

Procedure

The procedure included: (1) a session in which the general user data regarding web experience and typical online behavior patterns were assessed through a “user profile” questionnaire, (2) instructions, (3) a training with four sites, not included within the set of 26 experimental websites, (4) the experimental navigation session including 26 information seeking trials within the randomized set of experimental websites, and the explicit estimates of the information seeking time followed by a subjective assessment of the navigated website usability.

Written instructions were displayed on screen using a standard Microsoft PowerPoint (PPT) presentation. Each participant was tested individually. The participant was first given a short introduction to the lab setup and to the psychological measurements used throughout the experiment; then, s/he was instructed about the purpose of the experiment (i.e., investigating how people find products on SME sites) and informed that neither the search engine nor the browser tools were active to support navigation during the information seeking task.

Each experimental trial included the following ordered sequence of events (Fig. 2 depicts events from 3 to 8):

Figure 2 The information seeking task.

Temporal sequence of main events included in an information seeking trial of our experiment. The PPT slide on the left specifies the information goal “Search Shidosha scissors” (center) and the link to the home page of the Leader Cam company (bottom). The home page includes the relevant link “Collections” in the top horizontal bar. A click on “Collections” activates a subordinate page, which contains several clickable regions. The upper right contains the name and a representative picture of the Shidosha collection. Clicking on this region terminates the task and activates the third page (i.e., the information goal located two clicks away from the home page).

(1) the participant was informed that a cross would have been shown to him/her for about 15 s and that, at the end of the information seeking task, s/he should estimate the information seeking time (IST) using such a duration as a reference;

(2) a 30-px-wide green cross was displayed at the center of the white screen for about 15 s;

(3) a static PPT slide displayed black on white the items to be found, together with a brief purposive description of the website within which the participant was going to navigate through, and a hyperlink to the destination site;

(4) the participant read the text within the PPT slide aloud and the experimenter clarified all doubts about the goal raised by the participant;

(5) a white blank screen was displayed for about 500 ms after hyperlink selection;

(6) the home page of the selected website was displayed within the browser;

(7) the navigation was interrupted by the experimenter when the target link was selected from the correct target page, which was always one click away from the starting page;

(8) the participant provided a verbal estimate of the amount of time spent to get the required information;

(9) the SUS form was provided and the participant rated the amount of agreement with each of the 10 items of the questionnaire;

(10) the next trial followed.

The random sequences of 4 training sites and 26 experimental sites differed across participants. Each experimental session lasted about 90 min.

Results and Discussion

Data analysis

We analyzed three measures of web navigation proficiency. The individual information seeking time IST (taking as valid values those below 150 s, which led to the removal of 10 out of 520 trials), the duration judgments of the information seeking navigation EST, and the EST/IST ratio (i.e., duration judgment ratio). Following attentional models of experienced duration (Thomas & Weaver, 1975; Zakay & Block, 1995), the ratio of subjective to objective duration is a standard synthetic measure of the cognitive load involved in the task. Measuring it in our navigation task thus allowed us to afford a comparison between duration judgments made in conditions that entailed different levels of load due to the different duration of each navigation trial.

Our empirically grounded knowledge-based measure of complexity C was calculated for each website in order to investigate the main hypothesis that only part of web elements (i.e., those relevant for information seeking) affects the information seeking time. Following Eq. (1) we first solved the nonlinear curve-fitting problem of extracting the set of 5 weights (one for each web element) that best accounted for the obtained distribution of individual ISTs using a least-squares procedure constrained to yield positive real solutions. The best-fitting combination of weights was the following: ωhorizontal bar = 0.31; ωleft vertical bar = 0.12; ωright vertical bar = 0.001; ωfooter = 0.017; ωcanvas = 0.80.

Notably, such weights closely resemble the relative frequencies of occurrence of the 5 web elements within the entire set of 1945 FVG SME sites from which they were selected, as described by a second order polynomial fit (β1 = − 2.4; β2 = 2.9; intercept = 0.07; r2 = 0.99). This preliminary result supports the normative validity of our combination of weights. For each tested home page, we thus calculated its C value entering into Eq. (1) the corresponding NL, the presence/absence of each web element Ei, and the corresponding weights (see panel A of Fig. 1 for a plot of the model according to our weights). Individual synthetic self-assessment measures of website usability were extracted from SUS agreement ratings following the procedure indicated by Brooke (1996), yielding scores on a 0–100 scale.

We analyzed our indices of web navigation proficiency using a step-wise procedure that contrasted linear mixed-effect (lme) models of increasing complexity (Bates et al., 2014), depending on the number of fixed effects, modelled by our candidate continuous predictors (C, NL, NE and/or standard complexity indices: number of words, number of segregated graphical regions, overall number of links, number of images, file size of JPEG screenshot) and their meaningful combinations. Models were fitted by minimizing the restricted maximum likelihood criterion (Laird & Ware, 1982). The participants and the ATECO-based website categories were treated as random effects, to control for both the individual variability in the latency of task execution and its dependency on individual knowledge of the navigated domain of information. We followed Bates (2010) and used this statistical procedure to obtain two-tailed p-values by means of a likelihood ratio test based on χ2 statistics (for a discussion of advantages of a lme model over the more traditional mixed-model analysis of variance see Kliegl et al., 2010). We calculated type 3 like two-tailed p values using the Kenward-Rogers approximation for degrees-of-freedom implemented in KRmodcomp’s function, R Package pbkrtest. Among the indices that have been proposed as reliable measures of the predictive power and goodness of fit of lme models we selected the concordance correlation coefficient, rc, providing a measure of the degree of agreement between observed and predicted values, in the −1 to 1 range (Vonesh, Chinchilli & Pu, 1996). Post-hoc tests were performed using Welch two sample t-tests with unequal variance and Cohen’s d as a measure of significant effect size.

Knowledge-based complexity vs. NL and NE

In order to understand how the structural elements of a web page relevant for information seeking can determine the speed of web information search, we first analyzed the independent and conjoined effects of the number of web navigation elements, and of the number of links embedded in web navigation elements. Figure 3A shows the average information seeking time as a function of the number of links embedded in web navigation elements, averaged within three categories: small (M = 8.0, range = 6–12), medium (M = 16.2, range 13–21), large (M = 27.9, range 22–56), based on cutoffs at the 33rd and 66th percentiles. Values on the x-axis are also coded by bubble size, to facilitate the comparison with Fig. 3B. The number of web elements is color-coded as shown in the legend on top.

Figure 3 Predicting information seeking time.

(A) Mean information seeking time as a function of the average numerosity of links embedded in web elements: small (8.0), medium (16.2), large (27.9). The average number of links is also coded by bubble size. The three numerosities of web elements are color-coded (see legend on top). Error bars represent ±1 SE. (B) Plot of the same data as a function of our complexity measure C. (C) Plot of the average observed IST for each of the 26 experimental sites as a function of the IST predicted by the model in Eq. (2). Bubble size codes the number of links embedded in web elements and bubble color codes the three web element numerosities. The three representative sites discussed in Fig. 1A are evidenced by home pages (a, b, c): as predicted by C, the observed IST for site c was larger than for site b although site c included both a smaller number of links (smaller bubble) and a smaller number of navigation elements (orange vs. red). The blue line is the lme model regression line and the shaded region corresponds to ± standard error of the regression. The dotted line is the reference line standing for an optimal prediction of observed IST based on Eq. (2) (i.e., null intercept and unitary slope).

Consider home pages with 2 and 3 web elements. Participants spent a larger amount of time to find information within sites with a larger number of relevant web options; overall, information seeking time increased of about 6.3 s (t = − 5.32, df = 288.29, p = 0.000, d = 0.57) as the number of web options increased from small to medium, and further increased of about 2.6 s, though not significantly (t = − 1.37, df = 264.77, p = 0.07), as the number of web elements increased from medium to large. This is consistent with previous results showing that website complexity as modelled by the number of navigation choices is negatively correlated with navigation accuracy.

However, taking all web element conditions together, the present results do not provide evidence of a mere effect of the number of web options. Information seeking time was strongly dependent also on the number of web elements displayed on the page; it increased about 7.9 s as the number of web elements increased from 1 to 2 (t = − 7.6, df = 269.27, p = 0.000, d = 0.67) and kept increasing of about 3.0 s as the number of web elements increased from 2 to 3 (t = − 1.76, df = 319.18, one tailed, p = 0.04). A semantic-based approach to web information seeking behavior cannot predict the overall effect of web elements, given that only active web links should enter the competition for web selection. The likelihood of selecting the correct link should not be affected by those structural components of the artifact, like web navigation elements, that are both inactive (i.e., acting on them does not cause any variation in the status of the interface) and void of an explicit semantic meaning (e.g., being graphic elements).

These results corroborated H1, and are consistent with a model of information seeking time based on our knowledge-based account of home page complexity, C. This is shown in Fig. 3B, where we recoded the 6 conditions corresponding to different NL–NE combinations in term of C, and replotted the average information seeking time as a function of C. Information seeking time proportionally increased with C regardless of the number of web elements and the number of links. Our C measure indeed accounts for the overall trend of information search speed while accounting for both the lack of a significant difference between conditions with an equal number of links (i.e., largest number group) but different number of web elements (MN3 = 22.8 s ± 2.57 s vs. MN2 = 18.31 s ± 1.44 s; t = − 1.52, df = 123.17, p = 0.13), and the lack of a significant difference between conditions with equal number of web elements (NE = 2) but different number of links (Msmall = 15.9 s ± 1.59 s. Mmedium = 17.4 s ± 1.4 s; Welch t = − 0.74, df = 127.9, p = 0.46), which are both critical for an additive model independently weighting the effects of the number of links and elements.

This is confirmed by the results of the lme analysis testing the effects of web elements and web links, once the effect of C is controlled. In a first lme model, disregarding the effect of C, we thus asked how individual information seeking times were affected by the number of web elements and the number of links. In this model information seeking time resulted to be positively affected by the number of web elements (β = 5.42 ± 1.47, F1,426.25 = 13.7, p = 0.000) and the number of links (β = 0.66 ± 0.31, F1,429.68 = 4.2, p = 0.04), but not their interaction (β = − 0.21 ± 0.11, F1,428.28 = 3.32, p = 0.07). More interesting, however, was to repeat the same analysis including C as a third independent covariate so to control for its effects. Both main effects of web elements (β = − 6.76 ± 3.80, F1,425.13 = 2.31, p = 0.13) and web links (β = − 1.21 ± 0.63, F1,427.92 = 3.1, p = 0.08) became non significant when C was included in the model. In this second model, the likelihood of information seeking times was thus completely explained by C (β = 0.92 ± 0.27, F1,430.33 = 10.27, p = 0.001). In the present investigation, therefore, there is no evidence that the actual number of navigation choices and web elements per se contribute to the perceptual response beyond what C can explain.

Furthermore, no significance decrement of fit was found when contrasting this second lme model with a model with C as the only covariate (rc slightly decreases from 0.356 to 0.353; χ22=0.34, p = 0.84) with slope = 0.50 ± 0.07 and intercept = 8.87 ± 1.34. According to Paap & Roske-Hofstrand (1986) and our Eq. (2), the lme estimated slope stands for the individual search strategy that, being equal to 0.5, denotes an exhaustive search strategy consisting in reading all links embedded within relevant web elements, while the intercept stands for the information search latency.

As depicted in panel Fig. 3C, our model finely describes the metric of information seeking times obtained in our experiment as the best linear fit describing the relationship between predicted and average observed search times (r2 = 0.55; F1.24 = 29.58, p = 0.00) is a line with unitary slope (1.11 ± 0.20 vs. 1, t = 0.549, p = 0.58) and null intercept (−1.69 ± 3.52, t = − 0.48, p = 0.63). Surprisingly, as shown by the insets in Fig. 3C, the model also accounts for cases that are at odds with a semantic-based approach to web information seeking behavior and with the commonly held idea that information seeking time should be a monotonic function of the number of navigation choices; as predicted by C, site c required a larger information seeking time than site b (t = 1.9, df = 21.52, one tail p = 0.04, d = 0.61), although it included a dramatically smaller number of navigation choices (21 vs. 56 number of links, and 2 vs. 3 navigation elements; see Fig. 1A for details).

Knowledge-based complexity vs. standard complexity metrics

A causal mediation analysis using lme as mediator model types was performed to verify if, according to H2, a web party effect did occur in our study: to what extent the effects of multiple standard metrics of page complexity on information seeking time can be accounted for by their effects on C as a mediator, which in turn affects information seeking time? If a web party effect does occur, then information seeking time variability should be accounted for similarly by C as well as by standard metrics of web page complexity. The standard metrics we considered were the following: the overall number of links, the JPEG file size of the home page screenshot (bytes), the number of segregated graphical regions, the number of words, and the number of images. The implemented versions of mediation analysis used as default simulation type a quasi-Bayesian Monte Carlo method based on normal approximation (Imai, Keele & Tingley, 2010). We used White’s heteroskedasticity-consistent estimator for the covariance matrix from the sandwich package (Zeileis, 2006) and a bootstrapping method with 2000 re-samples to compute confidence intervals for the indirect effect, as well as—specifically—to determine whether the mediator completely or partially mediated the effect of predictor variables on the outcomes.

We summarized the result of the mediation in Fig. 4. The analysis allowed us to infer the Total Effect of multiple predictor variables on the IST(rightmost coefficients in Fig. 4), whether these predictors contribute to the variance of C as mediator (leftmost coefficients in Fig. 4), and to what extent the mediator contributes to the variance of the IST, Indirect Effect (top right coefficients in Fig. 4). Finally, the inferred Direct Effect (middle coefficients in Fig. 4) provided a measure of whether predictor variables continued to predict the IST with the mediator in the model.

Figure 4 Causal lme mediation analysis on information seeking time (IST) using standard indices of complexity as predictor variables and C as the mediator.

Color codes the different variables and their effects, with the mediator and the outcome variable in black, the Links × Segregated Graphical Regions interaction in grey and the 5 predictor variables in chromatic colors. Unstandardized estimates ±1 SEM are included in the model. Coefficients marked with two or three asterisks are significant at p < 0.001 or p < 0.0001 level. The effect of predictor variables on the C mediator variable is shown above the arrow lines connecting the 5 leftmost boxes and the grey outlined circle coding the Links × Segregated Graphical Regions interaction with the black outlined C box at the top of the model. The lme estimates of the Total Effects of the 6 predictor variables on IST are included in the righmost part of the model next to the IST black outlined box. The direct effects (with C as mediator) for IST are depicted above the arrow lines connecting 5 leftmost boxes and the grey outlined circle with the IST black outlined box. The proportion of effect mediated by C for the 6 predictor variables is depicted above the black arrow line connecting the C box to the IST box.

As regards the estimation of the Total Effect, we tested a first lme model inspired from previous works providing a compound measure of the artifact complexity based on a weighted linear combination of each single standard metric of web page complexity (Ivory, Sinha & Hearst, 2000; Ivory, Sinha & Hearst, 2001; Michailidou, Harper & Bechhofer, 2008; Harper, Michailidou & Stevens, 2009; Miniukovich & De Angeli, 2014). The simplest lme model accounting for the largest quote of variance (χ72=48.34, p = 0.000; rc = 0.354) amongst all candidate lme models combining all alternative metrics to C as fixed effects resulted to be one including significant main effects for the number of segregated graphical regions (F1,278 = 19.00, p = 0.000), the overall number of links (F1,426.6 = 9.76, p = 0.002), and the number of words (F1,426.5 = 8.15, p = 0.005), as well as a significant number of segregated graphical regions × overall number of links interaction (F1,419.4 = 8.65, p = 0.008). All predictor variables contributing to the variance of IST also contributed to the variance of the mediator C. The C value decreased as the number of words (β = − 0.11 ± 0.002; F1,423.2 = 26.02, p = 0.000) and the combination of links × graphical regions (β = − 0.013 ± 0.008; F1,417.9 = 229.30, p = 0.000) grew larger; while it increased with the JPEG file size (β = 0.012 ± 0.003; F1,411.8 = 12.59, p = 0.000), the number of links (β = 1.33 ± 0.068; F1,427.3 = 358.1, p = 0.000) and the number of segregated graphical regions (β = 4.27 ± 0.28; F1,235.1 = 191.7, p = 0.000).

However, consistently with the occurrence of a web party effect and with H2, when C was added as a covariate all effects of standard complexity metrics on IST became non significant, with their coefficients getting statistically equal to zero (middle coefficients in Fig. 4). All Total Effects were thus accounted for by C, demonstrating a full mediation of C of the effects of multiple metrics of page complexity on IST. All proportions of C mediated effects indeed resulted to be strongly significant: with C mediating about 31% the effects of the number of words, 109% the effect of the number of links, 53% the effect of the number of segregated graphical regions, and 101% the link × graphic interaction.

Again C resulted to be the only significant predictor of IST (β = 0.47 ± 0.11; F1,409.88 = 16.55, p = 0.000). Furthermore, no significant loss in the fit was found when contrasting an lme model with C as the only covariate vs. an lme model including all the effects of multiple metrics of page complexity (χ62=10.73, p = 0.09).

This result, together with the one discussed in the previous subsection, further demonstrates the occurrence of a web party effect. The web components of the home page accounting for the time to reach an information goal within a website are the links embedded within relevant web navigation elements. All other links and web objects do not affect search time, once the effect of our knowledge-based measure of home page complexity C is controlled.

Knowledge-based complexity and website usability

Previous results demonstrated that C has a strong impact on information search efficiency. According to H3, navigating within a low C site provides the user with a more rewarding experience, which should enhance the global experience of both pleasantness and usability. Following this hypothesis, we expected that C impacts the subjective estimates of the pleasantness of user-site interaction thus affecting website usability, with a larger effectiveness than alternative metrics of artifact complexity. To test this expectation, we compared C against users’ usability rating and contrasted its predictive power against that of other standard complexity metrics.

Usability scores were strongly predictive of information seeking time, which decreased at an lme estimated rate of about −0.389 ± 0.03 s every unit of the SUS scale (χ12=138.61, p = 0.000, rc = 0.54). Such a strong relationship demonstrated that usability experience, as measured through the SUS scale, well reflects the easiness of interaction which in turn is operationalized by the time needed to reach the information goal within a site.

In a further lme model we showed that usability ratings were finely predicted by our C index. SUS scores decreased at an estimated rate of about −0.66 ± 0.080 for every unit of C increment (χ12=58.9, p = 0.000, rc = 0.56). No improvement of fit was found if all other standard metrics, with the exception of the number of segregated graphical regions, were added to such a model (χ42=6.72, p = 0.15, rc = 0.56). The goodness of fit significantly improved when usability ratings were analyzed using an additive model with C and the number of segregated graphical regions as fixed effects (χ12=17.37, p = 0.000, rc = 0.58). As shown in Fig. 5, average ratings indeed decreased, at an almost constant rate, as the number of graphical regions increased, irrespective of whether C was small (Fig. 5A), medium (Fig. 5B), or large (Fig. 5C).

Figure 5 Usability judgments (SUS score) as a function of the number of segregated graphical regions, for three levels of C.

Average SUS scores [±SEM] for the 26 experimental sites as a function of the number of segregated graphical regions (NG), subdivided into three equal C intervals (A, B, and C panels include data for 4.9, 17.3, and 25.2 average C values, respectively). The home page complexity C is color-coded as shown in the vertical legend. The blue line is the lme model regression line and the shaded region corresponds to ± standard error of the regression. Horizontal dotted lines represent the grand average SUS scores [± SEM] for each C level (panel A → green; panel B → cyan; panel C → pink). The inset in (D) represents the average SUS score as a function of the average number of segregated graphical regions (NG), computed within three equal percentile intervals within the 3–10 range.

The net effect of the number of segregated graphical regions on usability judgments is depicted in Fig. 5D, showing that the SUS score significantly decreased as the average number of segregated graphical regions increased from 3.5 to 5.8 (83.5 ± 1.17 vs. 75.43 ± 1.28, t = 4.7, df = 409.40, p = 0.000, d = 0.45) as well as from 5.8 to 9.0 (75.43 ± 1.28 vs. 65.48 ± 2.35, t = 3.7, df = 156.9, p = 0.000, d = 0.48). The independent effect of C on SUS scores was supported by post-hoc analyses of usability judgments averaged across the three equal C intervals, each defining a panel of Fig. 5: websites were judged to be increasingly less usable as the average C increased from small (C = 4.9; green dotted line in Fig. 5A) to medium (C = 17.3; blue dotted line in Fig. 5B) (84.15 ± 1.84 vs. 75.63 ± 2.36, t = 4.67, df = 326.67, p = 0.000, d = 0.49), as well as from medium to large (C = 25.2; pink dotted line in Fig. 5C) (75.63 ± 2.36 vs. 68.27 ± 3.84, t = 3.13, df = 302.7, p = 0.001, d = 0.35).

In summary, the present results reveal that, consistently with H3, C is predictive of both information search efficiency and the subjective experience of pleasantness of user-website interaction, as represented by a usability judgment. Furthermore, we unexpectedly found that, once the effect of C is controlled, experienced usability is affected also by the number of segregated graphical regions. This unexpected effect is consistent with previous studies revealing a strong relation between the number of segregated graphical regions and the aesthetic appraisal of a webpage (Michailidou, Harper & Bechhofer, 2008). In our task, the subjective experience of pleasantness measured by the SUS might have been affected by the aesthetic components of the webpage, independent of C.

Knowledge-based complexity and cognitive load

Several studies have shown that time estimation is a reliable and valid measure of cognitive load, with experienced duration decreasing as the task gets increasingly difficult or attention-demanding (Block, George & Reed, 1980; Block & Zakay, 2008; Brown, 2008). These effects have been shown to be particularly strong under a time estimation paradigm similar to the one we used; i.e., under a prospective, rather than retrospective, paradigm in which the participant is aware, prior to the onset of the primary (information seeking) task that a duration judgment will be asked as a secondary task.

The rationale behind H4 arises from the influential scalar expectancy theory (Gibbon, 1977), which models prospective timing as a pacemaker-accumulator. According to such a theory we expected that, as the processing demands involved in the navigation task increase (as represented by C), experienced duration decreases, given that a larger part of attentional resources dedicated to the accumulator that counts the internal clock pulses should be invested in the primary task. Information seeking time is thus expected to be accounted for by a lme model including a significant effect of both C and estimated time. Such an expectation should be synthesized by the duration judgment ratio (i.e., ratio of subjective duration to objective duration), which should decrease as C increases.

Average information seeking times, shown in the top panels of Fig. 6 (panels A, B and C) together with the prediction of an lme model with both C and duration estimates as fixed effects, are in good agreement with our expectations (H4). Information seeking time proportionally increased with participants duration estimates (rc = 0.94) at an average rate of about 1.36 ± 0.025 s every estimated second (F1,474.8 = 2,764, p = 0.000), consistent with a global underestimation of duration congruent with previous results (Tractinsky & Meyer, 2001; Rau, Peng & Yang, 2006; Wood, Griffiths & Parke, 2007; Tobin & Grondin, 2009), with constant increments of about 0.756 ± 0.55 s, in small C sites relative to medium C sites, and 2.98 ± 0.57 s, in small C sites relative to large C sites (F1,375.2 = 13.87, p = 0.000). Such increments reflected the significant effect of C over duration judgments reveled by a no-intercept model with C as the only predictor, (β = 0.226 ± 0.047, F1,391.022 = 22.10, p = 0.000): as depicted by the colored dashed lines in Fig. 6, duration judgments increased by about 2.94 s (t = − 3.76, df = 354, p = 0.0002, d = 0.40) as C increased from small to medium, and by about 2.28 s (t = − 1.74, df = 330, p = 0.082, d = 0.20) as C increased from medium to large.

Figure 6 The duration judgment ratio depends on C.

(A, B, C) Average information seeking time [± SEM] as a function of the duration of information seeking [± SEM] for each of the 26 experimental websites, subdivided into the same three C intervals used in Fig. 5. The color of the dots codes the C value as described in the vertical legend. The blue line is the lme model regression line and the shaded region corresponds to ± standard error of the regression. Horizontal dotted lines represent the grand average estimated duration [± SEM] for each C level (panel A → green; panel B → cyan; panel C → pink). (D) Average duration judgment ratio as a function of C level (color-coded as the horizontal lines and shaded regions in the upper panels).

This result demonstrates that C is a determinant of duration judgments but does not provide direct evidence on how C affects the cognitive load involved in our information seeking task, given that in our experiment the duration of the task necessarily co-varied with the information-processing (attentional or working-memory) demands. In order to analyze the effects of C when the levels of cognitive load were controlled, we conducted a further analysis of duration judgment ratios. The pattern shown in the graph of Fig. 6D corroborated our expectation: duration judgment ratios decreased as C increased from small to large (0.69 ± 0.021 vs. 0.63 ± 0.023, t = 1.88, df = 330, p = 0.031, d = 0.30). Such effect is further confirmed by the results of a no-intercept lme model with C as the only predictor: the rate of decrement −0.0026 ± 0.001 was significant (χ12=5.25, p = 0.02, rc = .65).

These results provide an empirical and theoretical foundation for our C measure. We can conceive C as an intrinsic feature of a website, directly inferable from the starting web page, reflecting the extent to which information seeking will involve time-shared attentional, executive, or working-memory resources.

In summary, we obtained four findings: (a) consistently with H1, information seeking time was accounted for by only the subset of navigation elements whose effectiveness is formalized by C (a weighted combination of web options and elements available in the start page that, according to the user’s knowledge of layout conventions of websites, are more likely to be relevant for the achievement of information goals); (b) consistently with H2, and with the occurrence of a web party effect, all other elements and metrics of artifact complexity do not affect search time beyond what relevant elements can explain (full C mediation effect); (c) consistently with H3, the usability appraisal of the user-artifact interaction is similarly moulded by C, which in turn (d) reflects the cognitive load involved in the information search task (consistently with H4).

Conclusion

Navigation within websites poses intriguing problems about what do users do, and how do users decide what to do when searching information within a structured environment. The present study provides relevant insights to answer both questions.

Many website designs (implicitly) and the semantic-based approach to web information seeking behavior (explicitly) share the expectation that site search is faster when the home page of a website includes a smaller number of web options. Our study did not provide clear empirical support for such an expectation. Similar search times were indeed found for sites whose start page included a different number of selection choices but equal number of web navigation elements, while different search times were found for sites whose start page included an equal number of selection choices but a different number of web navigation elements. This is consistent with a web party effect as information seeking time in our experiment was dependent on part of (not all) web elements and links: those embedded within navigation elements that are more likely to be useful for information seeking. All other web elements (images, words, segregated graphical regions) and links do not contribute to information seeking time beyond what the part of web elements and links functional to web navigation can explain.

Such a result was accounted for by C, a composite measure of home page complexity based on the knowledge of layout conventions. C is an objective measure of the complexity of the start page, based on the number of links weighted by the number and type of embedding web elements. We indeed found that C fully mediated the effect of standard complexity metrics on information seeking time. Importantly, our C measure is consistent with a mechanism of selective attention that does not necessarily include an estimate of the relevance of all links within the web page. The predictive power of C revealed by our study demonstrates that participants can optimize the deployment of attentional resources necessary to navigation by focusing on links that are more likely to conduct to the information goal (i.e., those embedded within navigation elements). Following such a process, the likelihood that the correct link will be in the attended region is maximized through statistical knowledge, and wasting effort on attending to information patches that are statistically unlikely to conduct to the goal is minimized, so that selective attention will foster a fast and pure forward search.

Applied computational theory and research in human computer interaction have provided several techniques for the estimation of label relevance as based on alternative models of attention process in web navigation (Chi, Pirolli & Pitkow, 2000; Card et al., 2001; Katz & Byrne, 2003; Miller & Remington, 2004; Blackmon, Kitajima & Polson, 2005; Kaur & Hornof, 2005). These models are generally grounded on semantic and spatial factors. Semantic aspects refer to the similarity between the narrative description of user’s goal and labels meaning; spatial aspects refer to label positions expected on the basis of scanning habits (like reading direction). Here we showed that also the knowledge of web layout conventions matters, given that links belonging to web navigation elements that are more likely to be encountered within the home page (e.g., within the horizontal navigation bar) are weighted more heavily than links belonging to elements that are less likely to be encountered (e.g., right vertical menu).

This calls for an update of current semantic-based approaches to web information seeking behavior (Card et al., 2001; Katz & Byrne, 2003; Miller & Remington, 2004; Budiu et al., 2006; Wu & Miller, 2007), which should go beyond the narrative description of user’s goals and consider the user’s knowledge of web layout conventions. Our results indeed suggest that users exploit their knowledge about web layout conventions and search for labels in web elements consistent with their goals according to the implicit knowledge of web layout conventions.

Relative to the debate (Rieman, Young & Howes, 1996; Pirolli & Fu, 2003; Miller & Remington, 2004; Brumby & Howes, 2004) on whether labels are processed sequentially or hierarchically, heuristically or rationally, our results together with our composite measure of labels’ relevance call for an integrated approach. Web information search behavior could be modeled as a dynamic/recursive process where candidate labels collected during the parsing of the page (into web objects based on perceptual attributes such as closure, continuity, and similarity) compete for visual attention. The guidance of visual attention would depend, in its turn, by a complex interplay between top-down and bottom-up attentional processes, where both top-down factors (e.g., knowledge of the goal and of layout conventions) and bottom-up factors (e.g., the web page layout) contribute to the competition for web selection. Following Paap & Roske-Hofstrand (1986), the optimal fitting values of the parameter of our information seeking time model (Eq. (2)) standing for the search strategy (i.e., indicating the proportion of items that need to be read before the user terminates the search) denoted an exhaustive search strategy: participants in our experiment thus tend to read all of the links embedded within the web element most relevant to their goal before switching their attention to the next web element. This result has several close relatives with pioneering cognitive models of web navigation like the CoLiDeS (Kitajima, Blackmon & Polson, 2000) or SNIF-ACT (Fu & Pirolli, 2007), as it shows the need to integrate sequential and hierarchical processing of labels, to account for the variability of information search times over the dizziness of web layouts.

In sum, our C measure provides a convenient and general way to model the label relevance, being free from user-dependent parameters and requiring a minimal amount of knowledge about web information seeking behavior. Moreover, our C measure is consistent with an efficient search strategy that avoids the large computational effort necessary for a selection based on an assess-all strategy, as used by several web navigation models.

Our results also shed light on the factors best contributing to the usability and appraisal of a site on the basis of information included in the home page. This is a particularly relevant topic for web design, since visual complexity of the home page may play a decisive role in the formation of the first impression, as relating to experienced pleasure and arousal, thus being a crucial determinant of the choice to continue exploring a site on behalf of the user. Our C measure, being strongly correlated with estimated usability, proved to be diagnostic of how the artifact design, regardless of content, was judged to attract the user. Importantly, C provides a novel way to model artifact complexity and usability from the home page; a way that is rather simpler than most of current proposals for the quantitative analysis of several attributes of web page layout and composition, as well as their relation to usability. A C-based tool might be relevant to software dedicated to support universal usability (Shneiderman & Hochheiser, 2001), as we think that algorithms for automated usability assessment might incorporate it to improve their predictive power.

Finally, the obtained inverse relation between C and duration judgment ratio (i.e., ratio of subjective to objective task duration) suggests that C can be conceived as an easy way to quantify the cognitive load involved in searching an information item within a website. This finding is in line with the predictions of models of experienced duration (Zakay, 1993; Macar, Grondin & Casini, 1994; Zakay & Block, 1995; Block & Zakay, 1996; Brown, 1997), conceiving prospective timing as a dual task in which signals reflecting the passage of time are accumulated in a cognitive counter (Wearden, 2004). The smaller the amount of cognitive load involved in the primary navigation task (represented by C), the larger will be the amount of time signal processed, given that more attentional resources will be available for the secondary temporal estimation task. Our finding is relevant within the experienced duration literature, as to date most studies using prospective timing have failed to use long durations and tasks with an adequate degree of ecological validity. The present study instead assessed the effect of cognitive load on web navigation in a naturalistic environment (SME websites), achieving results that are consistent with recent evidence on gamers (Rau, Peng & Yang, 2006; Wood, Griffiths & Parke, 2007; Tobin, Bisson & Grondin, 2010): users underestimate interaction duration and the amount of underestimation proportionally increases as complexity (Sanders & Cairns, 2010) and interaction duration increase (Tobin & Grondin, 2009). Similar underestimation of duration and dependency of experienced duration on task complexity were found in participants searching for a target information within hierarchical menus (Tractinsky & Meyer, 2001).

Our work is in line with the recent proposal by Van Schaik & Ling (2012): it indeed supports the need for an integrated approach to the study of cognitive and experiential factors in HCI for the modeling of web navigation. Consistently with Van Schaik & Ling (2012), we found that cognitive and experiential factors together do indeed influence information search speed in web navigation. In particular, home page complexity has an effect on search speed, experienced complexity, usability, and cognitive load.

Additional Information and Declarations

Competing Interests

Author Contributions

Human Ethics

1 Enterprises with less than 250 employees and an annual turnover less than 50 million Euro (UE definition, L-124, 20 May 2003).

The authors declare there are no competing interests.

Sara Rigutti conceived and designed the experiments, performed the experiments, contributed reagents/materials/analysis tools, wrote the paper, reviewed drafts of the paper.

Carlo Fantoni conceived and designed the experiments, analyzed the data, wrote the paper, prepared figures and/or tables, reviewed drafts of the paper.

Walter Gerbino reviewed drafts of the paper, sample and data acquisition.

The following information was supplied relating to ethical approvals (i.e., approving body and any reference numbers):

1. Ethics Committee of the University of Trieste

2. EC UNITS, A.N: 52, 12.2.2013.

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
