# Peer review of "Web party effect: a cocktail party effect in the web environment"

_PeerJ, doi:10.7717/peerj.828_

## Round 0.1 · original submission · Minor Revisions

Please note that both authors thought your paper was interesting, but there are issues with the English. Please revise accordingly and have it carefully edited by somebody with excellent English writing skills.

·

Basic reporting

No Comments

Experimental design

No Comments

Validity of the findings

No Comments

Additional comments

The submitted article is a very well written paper that discuss an interesting and active field of research: navigational behavior of Web users. The authors have cited all the relevant literature from the HCI field and have compared their approach and measure with a plethora of other appraoches known from the literature. The experimental design is excellent and the results are interesting and novel.

Reviewer 2 ·

Basic reporting

I found this article to be sufficiently clear for me to understand the methods used and the claims made. I believe that the introduction and background were sufficient as well. That said, I think the paper could be slightly improved if the authors would have it read critically by a native English speaker. In addition, the figures were relevant and well presented, though I have some minor suggestions for improvement.

Here are some minor comments and questions about passages that confused me:

line 58: I believe "not the latest" should be "not the latter"

line 61: I gather that "Small Medium Enterprises" is a standard list of businesses known to those who work in this area, but it was not clear to me what this meant when it first appeared in the text, so I suggest the authors explain what this is here.

line 92: I suggest that the authors define what they mean by "long-term client-side" where it appears here.

line 109: What is meant by "physiology" in the context of this sentence?

line 180: I was surprised that the authors expected that C should be positively correlated with web-site usability EU; shouldn't C go down with greater "usability"?

line 193: I believe IST and EU were defined before they appeared on this line, but I did not notice a definition for EST (estimated search time?).

line 232: What is FVG?

line 316: "Each experimental session lasted about 1 hour and half minutes." Should this be "1 and a half hours"?

line 350: It would help if the authors gave a reference for "Restricted Maximum Likelihood" here.

line 379: "small for M = 8, medium for M = 16.2, and large for M = 27.9" does not seem to agree with the key labeled "number of web elements" at the top of the figure, which equates grey with 1, orange with 2, and red with 3. I am probably just confused about what the colors mean.

line 394: Is accuracy positively or negatively correlated with site complexity?

line 549: I was surprised to see in Fig. 5 that SUS score varied with NG once C was controlled for given claims made in the text, but I probably just misunderstood the detailed claims made.

Experimental design

I believe that the experimental design was appropriate and sufficiently rigorous to support the authors' claims.

Validity of the findings

I believe the findings were valid based on the evidence presented.

---

## Round 0.2 · accepted · Accept

Thank you for your careful and comprehensive revisions.